# The Association between Dietary Intake, Asthma, and PCOS in Women from the Australian Longitudinal Study on Women’s Health

**DOI:** 10.3390/jcm9010233

**Published:** 2020-01-15

**Authors:** Jessica A Grieger, Allison Hodge, Gita Mishra, Anju E Joham, Lisa J Moran

**Affiliations:** 1Robinson Research Institute, University of Adelaide, North Adelaide, SA 5005, Australia; Lisa.Moran@monash.edu; 2Adelaide Medical School, University of Adelaide, Adelaide, SA 5005, Australia; 3Cancer Epidemiology Division, Cancer Council Victoria, Melbourne, VCT 3004, Australia; Allison.Hodge@cancervic.org.au; 4Centre for Epidemiology and Biostatistics, Melbourne School of Population and Global Health, University of Melbourne, Melbourne, VCT 3010, Australia; 5School of Public Health, Faculty of Medicine, The University of Queensland, Brisbane, QLD 4072, Australia; g.mishra@sph.uq.edu.au; 6Monash Centre for Health Research and Implementation, School of Public Health and Preventive Medicine, Monash University, Clayton, VCT 3168, Australia; Anju.Joham@monash.edu; 7Diabetes and Vascular Medicine, Monash Health, Clayton, VCT 3168, Australia

**Keywords:** asthma, polycystic ovary syndrome, dietary patterns, diet, non-core foods, women

## Abstract

Dietary intake potentially modifies the prevalence or severity of asthma. The prevalence of asthma is higher in women with polycystic ovary syndrome (PCOS); it is not known if diet confounds or modifies the association between asthma and PCOS. The aims of this study were: (i) To determine if the association of PCOS and asthma is independent of dietary pattern and (ii) to determine if dietary pattern modifies the association between PCOS and asthma. Women in this study were from the Australian Longitudinal Study on Women’s Health (ALSWH) cohort born between 1973 to 1978 and aged 18 to 23 years (*n* = 7382). Logistic regression was used to assess the association between PCOS and asthma, adjusting for the following: (i) Potential confounders identified a priori and (ii) dietary patterns (*z*-score) identified by principle component analysis. In the adjusted analysis, women with PCOS were more likely to have asthma than the women without PCOS (OR 1.35 and 95% CI, 1.02 and 1.78). This relationship was not altered by further adjustment for dietary patterns (non-core food, meats and takeaway, or Mediterranean-style pattern). In the interaction analysis, only the women consuming less than the median intake of non-core foods (i.e., lower intake of discretionary or unhealthy foods) and with PCOS were more likely to have asthma (OR 1.91 and 95% CI, 1.29 and 2.82). Dietary intake did not confound the relationship between PCOS and asthma. Other mechanistic pathways are likely responsible for the asthma and PCOS association, and further studies assessing factors such as oral contraceptive use and sex steroid hormones warrant investigation.

## 1. Introduction

Polycystic ovary syndrome (PCOS) is an endocrine disorder with a reported overall prevalence rate of 6% to 10% [1] based on the Rotterdam diagnostic criteria of two of three features, i.e., oligo- or anovulation, hyperandrogenism (clinical or biochemical), and polycystic ovaries on ultrasound, after exclusion of other causes [2]. PCOS is associated with reproductive (hyperandrogenism, anovulation, infertility, and pregnancy complications) [3] and metabolic features (elevated prevalence of and risk factors for type 2 diabetes and cardiovascular disease) [4,5], along with psychological impact, such as increased risk of depression and anxiety [6] and impaired quality of life [7]. Insulin resistance is a key pathophysiological contributor to PCOS. Intrinsic insulin resistance is present in PCOS, with cases being more insulin resistant as compared with women of similar weight without PCOS [8]. Excess body weight exacerbates insulin resistance and the severity of PCOS symptoms [9]. Inflammation is also proposed as a pathophysiological contributor to PCOS [10]. This is not only as a consequence of elevated obesity, as women with PCOS have higher levels of proinflammatory markers such as C-reactive protein than BMI-matched controls [11,12,13]. Conversely, other studies have revealed that women who have PCOS do not have higher levels of inflammation than women without PCOS [14,15].

Asthma is a heterogeneous and complex disease, involving alterations in airway structure and airway inflammation and increased systemic inflammation [16]. It is one of the most common medical conditions with a prevalence of up to 38% in children [17] but only around 4% in adults [18]. Furthermore, a higher prevalence of asthma has been documented in women with PCOS from international studies including Danish [19,20] and Australian populations [21,22,23] which is hypothesized to be related to the elevated inflammation associated with PCOS. Diet can also modify the presentation of asthma through mechanisms including pro or anti-inflammatory effects. Poor dietary intake, such as foods high in saturated fats and glucose, contribute to inflammation through pathways involved in oxidative stress and insulin resistance, while foods high in antioxidants (such as fruits and vegetables) or unsaturated fatty acids (such as fish and nuts) are anti-inflammatory [24,25]. In keeping with these dietary effects on inflammation, a high fat/sugar/takeaway dietary pattern was associated with an increased likelihood of uncontrolled asthma in pregnancy [26]; and a high fat intake increased neutrophilic airway inflammation in subjects with asthma [27]. Higher dietary antioxidant intake was associated with improved lung function and asthma control (reviewed in [28]). In adults with asthma, those who consumed a low antioxidant diet for 14 days, were twice as likely to have an exacerbation at any time as compared with those consuming a high antioxidant diet [29].

It is unclear whether the association between asthma and PCOS is independent of diet given that both asthma and PCOS are associated with chronic low-grade inflammation. It is also unclear whether better dietary intake such as an anti-inflammatory diet could reduce the risk of asthma in women with PCOS. Determining this has important implications for understanding the pathophysiology of asthma in PCOS. We have previously examined dietary patterns in the Australian Longitudinal Study on Women’s Health (ALSWH) and identified a range of potentially anti-inflammatory (Mediterranean-style) and proinflammatory (non-core foods or meats and takeaway) dietary patterns [30]. The current analysis will build on this work to assess the following hypotheses: (i) Dietary intake confounds the relationship between asthma and PCOS, such that a more proinflammatory dietary pattern increases the risk of both PCOS and asthma and (ii) dietary intake is an effect modifier on the association between PCOS and asthma, such that following a healthier diet after a diagnosis of PCOS could reduce the risk of asthma.

## 2. Methods

### 2.1. Study Population

This study is based on ALSWH, a longitudinal population-based study of four age cohorts of Australian women. This analysis was conducted using data from women in the young cohort of ALSWH, aged 18 to 23 years at ALSWH commencement in 1996. Women were randomly selected from the national health insurance scheme (Medicare) database with intentional oversampling from rural and remote areas [31]. Further methodological details have been reported elsewhere [32,33]. The Human Research Ethics Committees of the University of Newcastle, Australia, and the University of Queensland, Australia, approved the study and informed written consent was obtained from each participant. This analysis comprised women at Survey 5 (in 2009) (aged 31 to 36 years), as PCOS status was only identified in Surveys 4 and 5, and food frequency questionnaire (FFQ) data was only collected in Surveys 3 and 5. We analyzed data from *n* = 7382 women who responded to the question on PCOS diagnosis in Survey 4 or 5 (responded yes to (”In the last three years have you been diagnosed with or treated for polycystic ovary syndrome”), who responded to the question on asthma at Survey 5 (”In the last three years have you been diagnosed with or treated for asthma”), and who had dietary data available for analysis (*n* = 550 PCOS and *n* = 6832 non-PCOS). No specific inclusion or exclusion criteria were applied, and all women were included, irrespective of pregnancy, medication, or country of birth. At Survey 5, the women sampled were broadly representative of the Australian population when compared with the census data [34].

### 2.2. Anthropometric and Demographic Variables

Information on self-reported demographic characteristics and anthropometric outcomes (height, weight, and waist circumference) were collected. Body mass index (BMI) was calculated using height and weight and respondents were categorized according to the World Health Organization guidelines for underweight (BMI < 18.5 kg/m^2^), normal weight (BMI 18.5 to 24.9 kg/m^2^), overweight (BMI 25 to 29.9 kg/m^2^), and obese (BMI ≥ 30 kg/m^2^) [35]. Demographic variables including country of birth, parity (no children or ≥1 child), education (no formal or high school education, diploma, trade, or degree), occupation (no paid job, clerical, trade, associated professional, or professional), personal income (no income, low income ($1 to 36,399 annually), medium income ($36,400 to 77,999 annually) and high income ($ ≥ 78,000 annually), marital status (never married, separated/divorced/widowed, or married/de facto), smoking status (no defined as never smoker or ex-smoker, or yes defined as any current smoking) and combined oral contraceptive pill use (yes/no) were collected at Survey 5 with the exception of country of birth which was obtained at Survey 1.

### 2.3. Dietary Intake Variables

Self-reported diet data were collected from the Dietary Questionnaire for Epidemiological Studies (DQES) Version 2. This is a FFQ developed by The Cancer Council of Victoria which was previously validated in young to middle-aged (16 to 48 year old) Australian women in comparison with a seven-day weighed food record [36]. Participants who had incomplete FFQ data (>10% of items with missing responses) or those who reported a daily energy intake of >14,700 kJ/day or <2100 kJ/day were excluded from the analyses. Intakes of 100 different foods (grams per day) were obtained from the FFQ and were assigned into 33 food groups (grams per day) for use in the dietary pattern analysis. As previously described [30], dietary patterns were derived using factor analysis with factor loadings extracted using the principal component method and varimax/orthogonal rotation. The number of dietary patterns identified was based on eigenvalues >1.5, on identification of a break point in the scree plot, and on interpretability [37]. Using these criteria, a three-factor solution was chosen and rerun with the resulting factor scores saved and converted to *z*-scores for analysis. Items with factor loadings ≥0.25 were considered as the items of relevance for the identified factor. These items represent the foods most highly related to the identified factor [38]. Foods that cross-loaded on several factors were retained.

### 2.4. Statistical Analysis

All statistical procedures were performed using Stata version 14.2 (StataCorp, 146 College Station, TX, USA). Frequencies and descriptive statistics were expressed as *n* (%) and as mean ± SD, respectively. All reported *p* values were two-tailed, and a *p* value <0.05 was considered to be statistically significant. Data were analyzed by independent *t*–test to compare continuous variables and Chi-square test to compare categorical variables between women with and without PCOS. Binary logistic regression analyses were used to assess the association between the dependent variable of asthma and the independent variable PCOS. Then, multivariable analyses were conducted adjusting for potential confounders identified a priori, including BMI, personal income, occupation, education, age, parity, smoking, marital status, country of birth, and oral contraceptive pill use. Additional multivariable analyses were then conducted adjusting for each dietary pattern (*z*-score) both in separate models for each *z*-score and in one model with all three *z*-scores included. Models were constructed to avoid collinearity and assessed for standard assumptions. Interaction effects between PCOS status and each dietary pattern (assessed as a categorical variable of above or below the median of the *z*-score) were assessed in the final models. Where interaction effects were significant, multivariable regression analyses were performed assessing the association between asthma as the dependent variable and PCOS status as the independent variable stratified for above and below the relevant median dietary pattern *z*-score.

## 3. Results

Participant characteristics are reported in Table 1. The prevalence of asthma was higher in women with PCOS as compared with those without PCOS (14.0 vs. 10.0%, *p* = 0.004). Women with PCOS had a higher BMI, weight and waist circumference, were less likely to have ≥1 child, less likely to be using the oral contraceptive pill, but more likely to have asthma.

The dietary intakes for women with and without PCOS are reported in Table 2. Women with PCOS had higher intakes of energy and fibre and a lower glycemic index, but differences were small.

Using principal components analysis, three dietary patterns were identified that explained 27% of the variance in food intake. These were non-core foods with high factor loadings for cakes, biscuits, sweet pastries; confectionary, refined grains and also takeaway foods and crisps; meats and takeaway with high factor loadings for fish (fried, processed, canned and cooked), processed meat, red meat, and takeaway food; and Mediterranean-style with high positive factor loadings for a variety of vegetables, fruit and nuts, and fish and inverse factor loadings for crisps. In an unadjusted analysis, women with PCOS had higher scores for all three dietary patterns, indicating a greater likelihood of following them.

In the unadjusted analysis, women with PCOS were more likely to have asthma than those without PCOS (OR 1.45 and 95% CI, 1.12 and 1.87) (Table 3). The association was retained in adjusted analysis (OR 1.35 and 95% CI, 1.02 and 1.78) (Table 3). When including the dietary patterns in separate logistic regression models, the relationship between asthma and PCOS was not altered. There was also no independent relationship between the non-core foods (OR 1.06° nd 95% CI, 0.97 and 1.17), meats and takeaway (OR 0.95 and 95% CI, 0.85 and 1.05), or Mediterranean-style dietary pattern scores (OR 0.94 and 95% CI, 0.86 to 1.03) and asthma in separate adjusted models.

The interaction between PCOS and the dietary pattern scores was assessed for asthma. There was no interaction between PCOS and the meats and takeaway (*p* = 0.449) or the Mediterranean-style dietary pattern (*p* = 0.917). There was an interaction between PCOS and the non-core foods dietary pattern (*p* = 0.015). There was an association between asthma and PCOS for women with a non-core foods dietary pattern *z*-score below the median (−0.15) (OR 1.91 and 95% CI, 1.29 and 2.82) but not above the median (OR 0.95 and 95% CI, 0.63 and 1.43). The relationship between asthma and PCOS was only observed for women with an overall healthier diet with lower intake of discretionary foods. The gram intake of food groups (cakes, confectionary, refined grains, vegemite, takeaway, crisps, juice, tomato sauce, processed meat, red meat, added sugar, and wholegrains) that scored ≥0.25 factor loading for the non-core foods dietary pattern are listed in Table 4 for the above and below median groups. Significant differences in median intake were found for all foods in the non-core foods dietary pattern, with the largest percentage difference for intake of cakes, confectionary, juice, crisps, and condiments including vegemite. 

## 4. Discussion

In a community cohort of Australian women aged 18 to 33 years, we confirm prior reports of a higher prevalence of asthma for women with PCOS than without, with a 1.35 higher likelihood of asthma with PCOS, or an absolute difference of 4% prevalence. We extend our previous research to also report that this higher prevalence of asthma was maintained after adjusting for dietary intake, specifically dietary patterns high in non-core foods, meats and takeaway, or a Mediterranean-style dietary pattern. These dietary patterns were not independently associated with asthma. However, we report for the first time an interaction between PCOS and the non-core foods dietary pattern score, such that the relationship between PCOS and asthma was only observed for women with a lower score on the non-core foods dietary pattern, indicating an overall healthier diet with lower intake of discretionary foods. 

Our results confirm the literature that has reported that PCOS is independently associated with asthma [20,23,39]. The relationship between asthma and PCOS is likely to be multifactorial, but there are mechanisms proposed that could explain the association. Higher circulating C-reactive protein, interleukin 6, and tumor necrosis factor alpha have been reported in PCOS which, through various mechanisms, can contribute to adipocyte dysfunction, insulin resistance, and type 2 diabetes [10,40]. Compared to women without PCOS, women with PCOS have higher free testosterone and ratio of free testosterone to free estradiol [41], and reproductive health issues including irregular menstrual periods and subfertility [7]. Similar to PCOS, asthma is proinflammatory [42]. Similar to women with PCOS, women with asthma more often have abnormal levels of progesterone and estradiol [43], experience irregular menstruation [44], and have a longer time to achieve a pregnancy [45] than women without asthma. Insulin resistance also has been shown to be a risk factor for allergic asthma in obese children and adolescents [46,47]. Although our study was based on self-reported information for PCOS and asthma, and we cannot disentangle the reproductive and metabolic disturbances for the different clinical PCOS phenotypes, common features of inflammation, insulin resistance, and altered hormones could have a role in the causation of asthma and allergy as well as in PCOS.

Unhealthy diets are associated with higher levels of inflammation [24,25]. Studies in adolescents or children have reported a higher risk for severe asthma in those consuming fast food three or more times per week [48] or with a low intake of fruits and vegetables, fish, butter, and dairy fat [49]. In adults, a high fish and fruit diet was related to a lower incidence of asthma [50]. Additionally, studies in adults with asthma report that those consuming higher fat and lower fiber intakes had lower forced expiratory volume in 1 s and airway eosinophilia [51]. A high-fat mixed meal increased sputum neutrophils 4 h post meal [27] but also resulted in increased airway inflammation and activation of a number of genes in sputum involved in immune system processes [52]. The literature, therefore, generally indicates that a healthier diet is associated with both a lower prevalence of asthma and better management of the disease. However, we did not find an association between any of the dietary pattern scores and asthma in this population. Controlling for these dietary patterns also did not alter the association between asthma and PCOS. These results suggest dietary intake does not confound the relationship between PCOS and asthma. Other exposures, such as OCP and sex steroid hormones, which are also associated with asthma [53] and PCOS [54,55], could play a larger role in the association between asthma and PCOS.

The literature is more contradictory with regards to the association between diet and PCOS. A number of studies have reported poorer dietary intake for women with PCOS including greater intakes of high glycemic index foods, fat, animal fat, saturated fat or cheese intake, and less fibre intake [56,57,58,59]. Conversely, we previously reported better diet quality, lower saturated fat and glycemic index, and higher fibre intake in women as compared with those without PCOS from ALSWH [60]. These discrepant findings may relate to a poorer quality diet pathophysiologically contributing to PCOS through weight gain or other mechanisms, such as worsening insulin resistance, or inflammation. Alternatively, a diagnosis of PCOS may lead to improved diet quality following health professional advice or motivation to change.

In this study, women with PCOS were more likely to follow all of the different dietary patterns with either a range of unhealthier foods in the non-core foods or meats and takeaway patterns consistent with a proinflammatory diet [24,25], or healthier foods in the Mediterranean-style dietary pattern consistent with an anti-inflammatory diet [61]. Interestingly however, we have conducted similar analyses using the Dietary Inflammatory Index as the dietary exposure but found no association with PCOS or asthma (unpublished). However, when we assessed the interactions between PCOS status and each dietary pattern, we found effect modification only for the non-core foods dietary pattern. Specifically, for women with a score below the median for this pattern, indicative of a healthier diet, the association between PCOS and asthma was observed. For women with a higher score there was no association between PCOS and asthma. These results are unexpected and lead us to reject our hypothesis that following a healthier diet after a diagnosis of PCOS reduces the risk of asthma. This result could indicate behavior change bias which can occur where women adopt behaviors perceived as healthier following diagnosis of a clinical condition such as asthma or PCOS, in association with clinician feedback or self-seeking of information for management. In this study, our prior findings of higher diet quality in community-based women with PCOS as compared with those without PCOS [30,62], may be further augmented by women with two clinical conditions, namely PCOS and asthma, having an even healthier diet higher in core foods and low in discretionary foods. This is consistent with the limited literature on dietary intake studies in adults with multiple chronic diseases where adults with type 2 diabetes and obesity had better diet quality than those with only type 2 diabetes [63]. 

The strengths of this study include the use of a large representative population-based cohort of young women. This allowed for the inclusion of varying degrees of obesity given that women with PCOS from clinical referrals have a higher prevalence of obesity than those from unselected populations [64]. Additionally, we considered a range of potential confounders in the relationship between PCOS and asthma. While the use of self-reported data for diet, asthma, PCOS, and anthropometry is a limitation, this is reasonable for use in a large epidemiological study. The FFQ has also been validated for use in young Australian women [36] and the self-reported BMI, as used in this study, has been validated with anthropometric measurement among the mid-age cohort from ALSWH [65]. Furthermore, we have shown that self-reported PCOS status is strongly associated with menstrual irregularity, a key diagnostic criteria for PCOS, i.e., the odds ratio for having PCOS was over 10 for women often reporting irregular menstrual cycles [66]. Limitations include that we did not assess lung function to confirm asthma status. However, it is unlikely there was misclassification bias as women would know whether they have asthma or not, and the diet exposure is unlikely to impact the probability of misclassification [67]. The absence of lung function data also meant we could not assess associations according to asthma severity. The cross-sectional nature of this study means we cannot infer causation, and thus longitudinal studies that can assess changes in modifiable risk factors, such as diet, could allow stronger inferences.

## 5. Conclusions

In conclusion, dietary intake did not confound the relationship between PCOS and asthma, and contrary to our hypothesis, a healthier dietary pattern did not lower the risk of asthma in PCOS but rather increased it. Although a complex relationship exists between dietary factors and risk for PCOS and asthma, our findings do not suggest that advising women with PCOS to consume more healthy diets is likely to help reduce the risk of asthma, but rather a healthy diet should be followed to support a range of general health outcomes. Other mechanistic pathways are likely responsible for the asthma and PCOS association, and further studies assessing factors, including oral contraceptive use and sex steroid hormones, warrant investigation. Further studies are also warranted to investigate if different clinical phenotypes of PCOS relate to asthma. Understanding if metabolic or reproductive features associate most with asthma would support treatment recommendations to reduce asthma prevalence.

## Figures and Tables

**Table 1 jcm-09-00233-t001:** Characteristics for women with and without polycystic ovary syndrome (PCOS).

	Non-PCOS *n* = 6832	PCOS *n* = 550	*p*-Value
Age (years)	33.7 ± 1.5	33.5 ± 1.4	0.005
BMI (kg/m^2^)	25.6 ± 5.8	28.5 ± 7.4	<0.001
Underweight	183 (2.8)	11 (2.1)	<0.001
Healthy weight	3532 (53.5)	205 (38.7)	
Overweight	1677 (25.4)	119 (22.5)	
Obese	1216 (18.4)	195 (36.8)	
Weight (kg)	70.8 ± 16.6	78.3 ± 21.6	<0.001
Waist circumference (cm)	86.2 ± 14.1	91.2 ± 17.2	<0.001
Smoking (yes)	989 (14.5)	74 (13.5)	0.436
Personal income			0.241
No income	604 (9.6)	55 (10.9)	
Low ($1–36,399/y)	2592 (41.1)	200 (39.5)	
Medium ($36,400–77,999/y)	2215 (35.2)	166 (32.8)	
High ($ ≥ 78,000/y)	890 (14.1)	85 (16.8)	
Education			0.658
No formal/high school	1398 (20.9)	104 (19.5)	
Trade/diploma	1790 (26.7)	140 (26.2)	
Degree	3513 (52.4)	290 (54.3)	
Occupation			0.788
No paid job	1425 (21.2)	108 (20.1)	
Clerical trade	1163 (17.3)	108 (20.1)	
Associate professional	1230 (18.3)	93 (17.3)	
Professional	2892 (43.1)	242 (50.0)	
Marital status			0.549
Never married	1178 (17.3)	103 (18.8)	
Separated/divorced/widowed	373 (5.5)	26 (4.7)	
Married/de facto	5262 (77.2)	419 (76.5)	
Parity (≥1)	4345 (63.6)	320 (58.2)	0.011
Born in Oceania	6437 (94.8)	510 (93.2)	0.116
OCP use (yes)	1581 (23.4)	96 (17.6)	0.002
Asthma	688 (10.0)	77 (14.0)	0.004

Data are presented as mean ± SD or *n* (%) and were analyzed by independent *t*-test (continuous variables) or Chi-square test (categorical variables). BMI, body mass index; OCP, oral contraceptive pill; and PCOS, polycystic ovary syndrome.

**Table 2 jcm-09-00233-t002:** Dietary intake in women with and without PCOS.

Dietary Component/Pattern	Non-PCOS *n* = 6832	PCOS *n* = 550	*p*-Value
Energy (kJ)	6746 ± 2262	6994 ± 2485	0.014
Carbohydrate (% energy)	40.3 ± 5.7	40.3 ± 5.8	0.963
Protein (% energy)	20.9 ± 3.3	21.1 ± 3.3	0.379
Fat (% energy)	37.0 ± 4.9	36.8 ± 5.1	0.566
Saturated fat (% energy)	15.4 ± 3.1	15.3 ± 3.2	0.273
Monounsaturated fat (% energy)	13.1 ± 2.1	13.1 ± 2.2	0.929
Polyunsaturated fat (% energy)	5.1 ± 1.6	5.1 ± 1.6	0.837
Glycemic load	86.9 ± 33.5	89.1 ± 35.7	0.136
Glycemic index	50.8 ± 4.0	50.4 ± 4.0	0.017
Fibre (grams)	19.1 ± 7.0	20.0 ± 7.9	0.002
Alcohol (grams)	9.4 ± 13.5	8.3 ± 13.3	0.081
Non-core foods dietary pattern	−0.03 ± 0.92	0.06 ± 1.00	0.035
Meats and take away dietary pattern	−0.04 ± 0.81	0.05 ± 0.89	0.008
Mediterranean style dietary pattern	−0.004 ± 0.99	0.12 ± 1.04	0.007

Data are presented as mean ± SD and were analyzed by independent *t*-test (continuous variables). PCOS, polycystic ovary syndrome.

**Table 3 jcm-09-00233-t003:** Logistic regression examining the association between PCOS and asthma.

Model	OR (95% CI)
Unadjusted	1.45 (1.12, 1.87)
Adjusted ^a^	1.35 (1.02, 1.78)
Adjusted ^a^, Non-core foods dietary pattern	1.34 (1.01, 1.78)
Adjusted ^a^, Meats and takeaway dietary pattern	1.35 (1.02, 1.79)
Adjusted ^a^, Mediterranean-style dietary pattern	1.36 (1.02, 1.80)
Adjusted ^a^, all three dietary patterns	1.36 (1.03, 1.80)

Data are presented as odds ratio, 95% confidence interval, and *p* values and were analyzed using logistic regression ^a^, Adjusted for BMI (WHO categories), income, occupation, education, age, parity, smoking, marital status, country of birth, and oral contraceptive pill use.

**Table 4 jcm-09-00233-t004:** Food group intake for the non-core foods dietary patterns.

Food	Factor Loading ^a^	Below Median (g/d)	Above Median (g/d)	Mean Difference (g/d), 95% CI ^b^	% Difference from below to above Median Intakes
Cakes, biscuits, sweet pastries (g/day)	0.661	10.2 ± 10.2	31.4 ± 29.0	−21.2 (−22.2, −20.2)	308% higher
Confectionary (g/day)	0.629	12.4 ± 12.0	32.6 ± 27.9	−20.2 (−21.2, −19.2)	263% higher
Refined grains (g/day)	0.483	74.8 ± 51.4	128.5 ± 78.1	−53.7 (−56.7, −50.6)	174% higher
Vegemite (g/day)	0.483	0.88 ± 1.1	2.2 ± 2.3	−1.3 (−1.4, −1.2)	250% higher
Takeaway (g/day)	0.467	30.5 ± 22.6	53.2 ± 37.5	−22.7 (−24.1, −21.3)	174% higher
Crisps (g/day)	0.466	14.1 ± 14.6	28.7 ± 26.6	−14.6 (−15.6, −13.6)	204% higher
Juice (g/day)	0.408	36.1 ± 54.5	90.3 ± 116.5	−54.3 (−58.4, −50.1)	250% higher
Tomato sauce (g/day)	0.380	1.8 ± 1.7	3.5 ± 3.5	−1.7 (−1.8, −1.6)	194% higher
Processed meat (g/day)	0.359	17.7 ± 16.2	28.5 ± 24.2	−10.9 (−11.8, −9.9)	161% higher
Red meat (g/day)	0.330	54.1 ± 44.2	76.7 ± 53.9	−22.6 (−24.8, −20.3)	142% higher
Added sugar (g/day)	0.325	8.8 ± 9.2	14.7 ± 12.8	−5.8 (−6.4, −5.3)	167% higher
Wholegrains (g/day)	0.319	74.9 ± 58.6	115.5 ± 90.7	−40.6 (−44.1, −37.1)	154% higher

^a^, Foods only included with factor loading >0.25 and ^b^, all *p* > 0.001 and were analyzed using independent *t-*test. Data are presented as mean ± SD or mean difference (95% confidence interval).

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
