# Peer review of "The Association between Dietary Intake, Asthma, and PCOS in Women from the Australian Longitudinal Study on Women’s Health"

_jcm, 2020, doi:10.3390/jcm9010233_

Round 1

Reviewer 1 Report

The article corresponds to the subjects and aims of Journal of Clinical Medicine.

The title and the abstract are written in a proper way and the abstract summarizes the content of the article. The aims were to determine whether the association of PCOS and asthma is independent of dietary pattern and whether dietary pattern modifies the association between PCOS and asthma. The article is very interesting, demonstrate by originality. Research results are also practical aspect.

The interpretation of the results is clearly presented and adequately supported by the evidence adduced.

Sentence: The prevalence of asthma was higher in women with PCOS compared to those without PCOS (14.0 vs 10.0%, p = 0.004) (line 169-170) should be moved to the description of table 1.

The conclusions are logically valid and justified by the evidence adduced. The quoted literature is selected properly for the subject of the article, and it is current enough. The article is possess scientific and practical value.

Author Response

Reviewer 1:

Comment 1: The article corresponds to the subjects and aims of Journal of Clinical Medicine.

The title and the abstract are written in a proper way and the abstract summarizes the content of the article. The aims were to determine whether the association of PCOS and asthma is independent of dietary pattern and whether dietary pattern modifies the association between PCOS and asthma. The article is very interesting, demonstrate by originality. Research results are also practical aspect.

The interpretation of the results is clearly presented and adequately supported by the evidence adduced.

Sentence: The prevalence of asthma was higher in women with PCOS compared to those without PCOS (14.0 vs 10.0%, p = 0.004) (line 169-170) should be moved to the description of table 1.

The conclusions are logically valid and justified by the evidence adduced. The quoted literature is selected properly for the subject of the article, and it is current enough. The article is possess scientific and practical value. 

Response 1: Thank you for your positive comments. We have made the change regarding moving the prevalence of asthma up to Table 1 description.

Reviewer 2 Report

The article is interesting, it is well written. The number of participants (n=7382) is very high. The statistical analysis is very complete. Many covariates have been included. However, there are some aspects that, in my view, need clarification. They are the following:

1. Introduction. The relationship between PCOS and inflammation is controversial, as it is shown in our article ("Are there differences in basal thrombophilias and C-reactive protein between women with or without PCOS?" published by Sánchez-Ferrer ML, et al. Reprod Biomed Online. 2019.

The authors present a clear relationship between PCOS and inflammation, but these association remains unclear, and this should be well established in the introduction.

2.Material and  method. Diagnosis of PCOS  by history  is not accurate as it is not based on Rotterdam criteria. Our experience is that there are differences between the prevalence of PCOS diagnosed made by history versus the diagnosis of PCOS made by applying thr Rotterdam criteria. Would it be possible with the data collected to make the diagnosis of PCOS applying this criteria? I think it would give validity to the article.

3. Result:  it is rare for PCOS  to take less contraceptives when it is the most frequent treatment for PCOS condition. Can the authors explain why the had these results?

4. Discussion and conclusion. It is not clear the association etiological  between PCOS and asthma and also it is not clear why  PCOS  and asthma women have a better diet .Could the authors provide a better explanation for both questions?

5. It would have been interesting to see what happens with different phenotypes of PCOS. All phenotypes of PCOS are related to asthma, only the anovulatory types or those with worse metabolic profile? Due you have a great number of women, it could be interesting to explore this question.

Author Response

Reviewer 2:

The article is interesting, it is well written. The number of participants (n=7382) is very high. The statistical analysis is very complete. Many covariates have been included. However, there are some aspects that, in my view, need clarification. They are the following:

Comment 1Introduction. The relationship between PCOS and inflammation is controversial, as it is shown in our article ("Are there differences in basal thrombophilias and C-reactive protein between women with or without PCOS?" published by Sánchez-Ferrer ML, et al. Reprod Biomed Online. 2019.

The authors present a clear relationship between PCOS and inflammation, but these association remains unclear, and this should be well established in the introduction.

Response 1: Thankyou for this comment. We now include in the introduction (line 50-52): “Conversely, other studies have also revealed that women who have PCOS do not have higher levels of inflammation than women without PCOS (Sánchez-Ferrer ML, et al. Reprod Biomed Online. 2019; Åsa Lindholm, Human repord 2011 p 476).

Comment 1. Material and method. Diagnosis of PCOS  by history  is not accurate as it is not based on Rotterdam criteria. Our experience is that there are differences between the prevalence of PCOS diagnosed made by history versus the diagnosis of PCOS made by applying thr Rotterdam criteria. Would it be possible with the data collected to make the diagnosis of PCOS applying this criteria? I think it would give validity to the article.

Response 2: We agree that the self-reported diagnosis of PCOS is a limitation of this study. However, there is no further data available on the different diagnostic features of PCOS in this dataset with the exception of menstrual irregularity. As stated on line 281, we have previously reported in survey 4 of the ALSWH that self-reported PCOS status was strongly associated with menstrual irregularity [64]. We have expanded this sentence to add that “the odds ratio for having PCOS was over 10 for women often reporting irregular menstrual cycles”.

Comment 3. Result:  it is rare for PCOS  to take less contraceptives when it is the most frequent treatment for PCOS condition. Can the authors explain why the had these results?

Response 3:

We have previously reported that women with PCOS in this age group were less likely to be using the oral contraceptive pill and more likely to be trying to conceive (https://www.ncbi.nlm.nih.gov/pubmed/24549213). The lower use of contraceptives here is therefore likely related to women ceasing these to attempt to conceive. It may also be related to women with PCOS being less concerned about unprotected sex as they consider their likelihood of unplanned pregnancy to be low (Jones GL,  Hall JM,  Lashen HL,  Balen AH,  Ledger WL. 

Health-related quality of life among adolescents with polycystic ovary syndrome, 

J Obstet Gynecol Neonatal Nurs, 2011, vol. 40 (pg. 577-588).

Comment 4Discussion and conclusion. It is not clear the association etiological between PCOS and asthma and also it is not clear why PCOS and asthma women have a better diet. Could the authors provide a better explanation for both questions?

Response 4:

i) It is difficult to disentangle the etiological association between PCOS and asthma because the data was based on self-reported information, thus we do not know whether reproductive or metabolic features were key contributing pathways. In our discussion, we stated there were common metabolic, inflammatory and reproductive features in both asthma and PCOS. We have also now added to the discussion, line 230-232 to state: “Although our study was based on self-reported information for PCOS and asthma, and we cannot disentangle the reproductive and metabolic disturbances for the different clinical PCOS phenotypes, common features of inflammation, insulin resistance and altered hormones may have a role in the causation of asthma and allergy as well as in PCOS.”

ii) Regarding the comment above and that women with PCOS and asthma have a better diet: We stated in our discussion that our results are unexpected and lead us to reject our hypothesis that following a healthier diet after a diagnosis of PCOS reduces the risk of asthma. We also elaborated on this result and suggested reasons including behaviour change bias; the impact of clinician feedback; and we reported on a previous study in adults with 2 chronic conditions who also had a healthier diet. Because this is the first study of its kind, we cannot speculate any further as to our results, and further clinical and mechanistic studies on this would be helpful.

Comment 5. It would have been interesting to see what happens with different phenotypes of PCOS. All phenotypes of PCOS are related to asthma, only the anovulatory types or those with worse metabolic profile? Due you have a great number of women, it could be interesting to explore this question.

Response 5:

We agree that further exploration of these findings according to phenotype of PCOS would be very interesting. However, there is no further data available on the different phenotypes of PCOS in this dataset. We now include the following in the conclusion, line 304-307: “Further studies are also warranted investigating whether different clinical phenotypes of PCOS relate to asthma. Understanding whether metabolic or reproductive features associate most with asthma will support treatment recommendations to reduce asthma prevalence.”

Round 2

Reviewer 2 Report

Thank you for your answers and changes in the text.